# Association between Ambient Temperature and Severe Diarrhoea in the National Capital Region, Philippines

**DOI:** 10.3390/ijerph18158191

**Published:** 2021-08-02

**Authors:** Paul L. C. Chua, Chris Fook Sheng Ng, Adovich S. Rivera, Eumelia P. Salva, Miguel Antonio Salazar, Veronika Huber, Masahiro Hashizume

**Affiliations:** 1Department of Global Health, School of Tropical Medicine and Global Health, Nagasaki University, 1 Chome-12-4 Sakamoto, Nagasaki 852-8102, Japan; chrisng@nagasaki-u.ac.jp (C.F.S.N.); hashizume@m.u-tokyo.ac.jp (M.H.); 2Department of Global Health Policy, Graduate School of Medicine, The University of Tokyo, 7-3-1, Hongo, Bunkyo-ku, Tokyo 113-0033, Japan; 3Alliance for Improving Health Outcomes, Inc., Rm. 406, Veria I Bldg., 62 West Avenue, Barangay West Triangle, Quezon City 1104, Philippines; mikesalazar@gmail.com; 4Institute for Public Health and Management, Feinberg School of Medicine, Northwestern University, 633 N. St. Clair Street, 20th Floor, Chicago, IL 60611, USA; AdovichRivera2021@u.northwestern.edu; 5San Lazaro Hospital, Quiricada St., Santa Cruz, Manila 1003, Philippines; dobymel@hotmail.com; 6Institute of Global Health, University of Heidelberg, Im Neuenheimer Feld 324, 69120 Heidelberg, Germany; 7Department of Physical, Chemical and Natural Systems, Universidad Pablo de Olavide, Ctra Utrera km 1, 41013 Sevilla, Spain; vehub@upo.es

**Keywords:** diarrhoea, hospital admissions, mortality, ambient temperature

## Abstract

Epidemiological studies have quantified the association between ambient temperature and diarrhoea. However, to our knowledge, no study has quantified the temperature association for severe diarrhoea cases. In this study, we quantified the association between mean temperature and two severe diarrhoea outcomes, which were mortality and hospital admissions accompanied with dehydration and/or co-morbidities. Using a 12-year dataset of three urban districts of the National Capital Region, Philippines, we modelled the non-linear association between weekly temperatures and weekly severe diarrhoea cases using a two-stage time series analysis. We computed the relative risks at the 95th (30.4 °C) and 5th percentiles (25.8 °C) of temperatures using minimum risk temperatures (MRTs) as the reference to quantify the association with high- and low-temperatures, respectively. The shapes of the cumulative associations were generally J-shaped with greater associations towards high temperatures. Mortality risks were found to increase by 53.3% [95% confidence interval (CI): 29.4%; 81.7%)] at 95th percentile of weekly mean temperatures compared with the MRT (28.2 °C). Similarly, the risk of hospitalised severe diarrhoea increased by 27.1% (95% CI: 0.7%; 60.4%) at 95th percentile in mean weekly temperatures compared with the MRT (28.6 °C). With the increased risk of severe diarrhoea cases under high ambient temperature, there may be a need to strengthen primary healthcare services and sustain the improvements made in water, sanitation, and hygiene, particularly in poor communities.

## 1. Introduction

Diarrhoeal diseases are a global health problem that cause a significant burden in the general population, especially in young children [1]. It was estimated that they caused 74 million disability-adjusted life years (DALYs) and 1.7 million deaths globally in 2016 [1]. With the threat of climate change, the incidence of diarrhoeal diseases is expected to increase due to the rising ambient temperature [2]. Numerous studies have reported an increased risk in diarrhoeal diseases brought about by warm temperatures [3]. Studies have explained that this occurs due to the adverse changes that warm temperatures bring to the environment, water and sanitation, as well as human behaviour, which contribute to the survival and transmission of enteropathogens [2].

Diarrhoea is generally self-limiting and resolves on its own [4]. Severe cases, on the other hand, require medical intervention due to dehydration, complications brought about by the inflammation of the intestinal mucosa, and serious co-morbidities [5]. These may explain the higher risk of fatality in severe diarrhoea cases compared to the less severe ones [6]. Such cases may be aggravated by high ambient temperature possibly due to the increased heat stress and reduced availability of clean drinking water, especially in impoverished conditions. This hypothesis has not been explored yet since many studies were limited to analysing the ambient temperature association in reported diarrhoea cases from primary healthcare facilities or outpatient visits [2,3]. Although some have analysed hospital admissions and mortality due to diarrhoeal diseases [7,8,9,10], none has specifically quantified the association between ambient temperature and severe diarrhoea cases.

In this study, we examined the association between ambient temperature and severe cases of diarrhoea in the National Capital Region (NCR) of the Philippines. We selected two health endpoints to represent severe diarrhoea cases: (1) Hospital admissions at a major referral facility since they can receive cases with considerable complications that require specialised treatment, which regular hospitals cannot provide, and (2) mortality with diarrhoea as the primary cause of death. The findings of this study could contribute to the development of policies for health adaptation and prevention of severe cases of diarrhoeal diseases.

## 2. Materials and Methods

### 2.1. Study Site

The NCR or Metropolitan Manila is the center of government, population, and economy of the Philippines (Figure 1). It comprises four districts (i.e., Capital, Eastern Manila, Northern Manila, and Southern Manila districts), and is further divided into 16 cities and one municipality with a total land area of 619.57 km^2^. It is the most densely populated administrative region with 20,785 individuals per square kilometer in 2015 [11]. The annual poverty incidence was 2.3% in 2018, which declined from 4.1% in 2015 [12]. The region has a tropical climate with relatively high temperature, humidity, and rainfall [13]. Rainy and dry seasons are the two main seasons in the country including the NCR [14]. The rainy season is from June to November, and the dry season is from December to May. The summer or hot dry season is from March to May. We selected three districts (i.e., Capital, Eastern Manila, and Southern Manila districts) with available weather data.

### 2.2. Data Sources

Individual death records of NCR residents were collected from the Philippine Statistics Authority with a date of death within 1 January 2006 to 13 December 2017. Additionally, the causes of death were coded A00 to A09 for intestinal infectious diseases based on the International Classification of Diseases (ICD) 10. We selected San Lazaro Hospital (SLH) in Manila City as a source of severe diarrhoea cases since it is a national tertiary special hospital, which serves as a referral hospital for infectious and tropical diseases for the entire country. It receives cases of diarrhoea with some degree of dehydration or serious complications. We collected data on hospital admissions of NCR residents from SLH with an admission date within 1 January 2006 to 13 December 2017 and ICD-10 code A00 to A09 based on the final diagnosis. We selected hospital admissions with dehydration and/or co-morbidity as written in the final diagnosis. Hospital admissions with final diagnosis of only infectious diarrhoea or gastroenteritis were excluded since they were presumed as non-severe cases. We separately aggregated the number of hospital admissions and deaths into a weekly time series by districts based on residential addresses.

We identified a weather station in each district (Figure 1) and extracted the daily mean temperatures, dew point temperatures, and total rainfall from the database of the Global Historical Climatology Network provided by the National Centers for Environmental Information, National Oceanic and Atmospheric Administration [15]. Since there is no meteorological station in the Northern Manila district, we did not include that district in the analysis. The proportion of days with missing temperatures and dew point temperatures was less than 1% for all the weather stations. We inputted the missing temperatures and dew point temperatures using multiple linear regression [16]. However, we were unable to include rainfall data due to the paucity of available data, 22% of the consecutively missing data at the weather station are in the Southern Manila district. Moreover, we computed the weekly average of daily mean and dew point temperatures by weather stations.

### 2.3. Statistical Analysis

We applied a two-stage time series approach to model the non-linear temperature associations of severe diarrhoea outcomes. In the first stage, we estimated the association between weekly average temperatures and weekly severe diarrhoea outcomes by fitting the generalised linear model in combination with a distributed lag nonlinear model [17]. We applied a quasi-Poisson link function to account for overdispersion in the count health outcome data. The model is given as:$$ln\left[E\left(D_{t,c}\right)\right]=\alpha +cb\left(temp_{t}\right)+ns\left(dewp_{t}\right)+ns\left(t\right)+holiday_{t}$$
where E(D) is the expected diarrhoea counts in week t for diarrhoeal outcome c, α is the intercept, cb(temp) is a cross-basis matrix for the exposure- and lag-response functions computed using the weekly average of mean temperatures, ns(dewp) is a natural cubic B-spline (*ns*) of dew point temperature with 3 degrees of freedom (*df*) included as a confounder [18], ns(t) is *ns* of week with 3 *df* per year for 12 years to control for the effects of seasonality and long-term trends, and holiday is the number of national holidays in a given week t.

We modelled the exposure-response function using a *ns* with two knots placed at the 33rd and 67th percentiles of the temperature distribution [19]. To account for the delayed effects of temperature on severe diarrhoea cases, we included temperatures in week 0–4 considering the complex pathological pathways involved as done previously [20]. The lag-response function was modelled using a *ns* with an intercept and three internal knots placed at equally spaced values in the log scale. The *df* for ns(t) was selected to minimise the sum of absolute values of the partial autocorrelation function of the residuals (Appendix A).

In the second stage, we used an intercept-only meta-regression model to combine the district-specific estimates [21]. Residual heterogeneity was measured using *I*^2^ statistics and Cochran Q test. We derived the best linear unbiased prediction (BLUP) of the overall cumulative temperature-severe diarrhoea associations presented as estimates of relative risks (RR) for each district. The BLUP represented a trade-off between district-specific and district-pooled associations enabling areas with a small number of daily deaths and hospital admissions to use information from areas with larger populations [21]. We computed the RRs for exposure to high- and low-temperatures by comparing the risk at the 95th and 5th percentiles of temperature distribution, respectively, to the risk at minimum risk temperature (MRT) selected as the reference. MRT was district specific, estimated as the lowest cumulative RRs restricted between the 1st and 99th percentiles temperature distribution [22]. The reference MRT was used to re-center the cumulative associations. We derived the overall NCR-level association using daily temperatures of all districts from the intercept-only model [23]. We examined the lag structure of the association for possible delayed effects by estimating the association at each lag from week 0–4 using the same percentile cut-offs for high-and low-temperatures and a reference temperature based on the regional average MRT.

### 2.4. Sensitivity Analysis

We examined the sensitivity of the cumulative associations to higher *df* for week and longer lag weeks [18]. We particularly checked the shape of the cumulative associations for mortality outcome by extending the lag to 7 weeks in view of reports of deaths that occurred up to 21 days after visiting a health center with diagnosis of diarrhoea [6]. Since any hospital admissions due to diarrhoea were defined as moderate-severe cases in another study [24], we checked the shape of cumulative associations for all SLH hospital admissions due to diarrhoea including non-severe ones. We also checked the cumulative associations of models that controlled for rainfall in Capital and Eastern Manila districts to see if rainfall was a confounder.

### 2.5. Software

All the analyses were performed in R statistical programming v4.0.4 using *dlnm* and *mixmeta* packages [17,25,26].

## 3. Results

There was a total of 5186 deaths due to diarrhoea in 2006–2017 in the three NCR districts. For hospital admissions, there were 75% or 5668 severe diarrhoea cases (i.e., cases with dehydration and/or co-morbidity) out of the total 7566 admissions due to diarrhoea in the same year and districts. Most of the deaths and hospitalised severe cases were residents from Eastern Manila and the Capital District, respectively (Table 1). The majority of the mortality and hospitalised severe cases were also pathogen-unspecific infectious gastroenteritis (Appendix A). Sixty-one percent of the mortality were attended by a physician, while 31% were not attended by any physician (Appendix A). More than half of mortality and hospitalised severe cases were under 5 years old (Appendix A). There was more proportion of 65 years and older in mortality than hospitalised severe cases. The average weekly mean and dew point temperatures in the region were 28.0 and 23.5 °C, respectively. Distributions of weekly mean and dew point temperatures were similar across the districts.

The cumulative associations of mean temperature with mortality and hospitalised severe cases were approximately J-shaped with a steeper upward trend at high temperatures, more so in the case of mortality outcome (Figure 2 and Appendix A). The estimated MRT for the NCR was approximately 28.2 °C for mortality and 28.6 °C for hospitalised severe cases. At high temperatures, the risk of mortality increased by 53.3% [95% confidence interval (CI): 29.4%; 81.7%] (Table 2). A similar increase was observed in hospitalised severe cases albeit a smaller one at 27.1% (95% CI: 0.7%; 60.4%). Mortality risk increased at low mean temperatures by 22.6% (95% CI: 0.4%; 49.6%). There was no evidence indicating the increased risk of hospitalised severe cases at low temperatures. We found no evidence of heterogeneity for the pooled estimates for mortality (*I*^2^ = 0%; Cochran Q = 5.41) and hospitalised severe cases (*I*^2^ = 0%; Cochran Q = 4.56).

Mortality risk at high temperature (95th percentile) was the highest at lag week 0 but quickly reduced to null thereafter (Appendix A). On the other hand, mortality risk at low temperature (5th percentile) appeared to be delayed and displayed a peak at lag week 2. For hospitalised severe cases, the risk at high temperature appeared to have two peaks at lag weeks 1 and 3. At low temperature, the pattern was more monotonous compared to the risk of mortality. Heterogeneity of the estimated associations were low for both severe diarrhoea outcomes (Appendix A).

Results of the sensitivity analysis showed that the overall shapes of cumulative temperature-mortality associations remained relatively unchanged when the *df* for week (i.e., the *df* defining the spline function to control for seasonal and long-term trends) and the number of lag weeks were increased, although at low temperature there was more variability (Appendix A). In terms of delayed effects, extending the lag to 7 weeks did not lead to a substantive change in the overall curve for mortality at high temperatures but did dampen the association at low temperatures. For hospitalised severe cases, the overall associations were more sensitive to the changes in the *df* for week and the number of lags (Appendix A). For all hospital admissions due to all diarrhoea cases (including non-severe cases), the cumulative associations with temperature showed an upward but insignificant slope at high temperatures (Appendix A). The shapes of cumulative associations with models controlling for rainfall were similar with the models without controlling for rainfall (Appendix A).

## 4. Discussion

We quantified the association between temperature and severe diarrhoea cases (in the form of mortality and hospitalised severe cases) for the first time in three districts of National Capital Region, Philippines. We found that both mortality and hospitalised severe cases had a strong evidence of association at high temperatures while less apparent at low temperatures, especially for hospitalised severe cases. Under high temperatures, risks of mortality appeared to be higher than risks of hospitalisation due to severe diarrhoea.

The temperature associations found may be explained by the aetiologies that comprise the severe diarrhoea cases. Under high temperatures, risks for shigellosis, *E. coli* enteritis, salmonellosis, and cryptosporidiosis (including other bacterial enteropathogens) increase while risks for rotaviral and noroviral enteritis increase under low temperatures [3,27]. Although an aetiology-specific analysis was not possible in this study, there are separate studies that determined aetiologies of diarrhoea in the Philippines. Adkins et al. reported that hospital admissions due to diarrhoea from San Lazaro Hospital in 1983–1984 were caused mostly by rotavirus, *Shigella* spp., *Salmonella* spp., and *Escherichia coli* [28]. In a more recent study from the Global Rotavirus Surveillance Network, the most common enteropathogens found in Philippine children under 5 years old with acute watery diarrhoea were rotavirus, norovirus, *Cryptosporidium* spp., *Shigella* spp., and *E. coli* [29]. For mortality due to diarrhoea, the findings from the Global Enteric Multicentre Study (GEMS) conducted in low-middle income countries suggest that the resulting deaths from moderate-severe diarrhoea (MSD) in children were primarily associated with *Escherichia coli* and *Shigella* spp. despite the fact that the majority of the MSD were due to rotavirus [6,30]. This may provide some explanation why high temperature associations were evident for mortality and hospitalised severe cases. Moreover, typhoid fever and amoebiasis, which were the most reported aetiology in both mortality and hospitalised severe cases, are found to be positively associated with ambient temperature [31,32,33,34]. The severe diarrhoea risks in low temperatures may be related to the viral aetiologies that were found to be common in children [29].

The increased susceptibility to severe diarrhoea under high temperatures can also be explained by dehydration and co-morbidity [5]. Severe dehydration due to uncompensated water loss from diarrhoea can compromise the body’s thermoregulation and electrolyte balance, which can be life threatening [35]. Children, elderly, and adults with chronic conditions are more susceptible to dehydration than normal adults due to their lower rate of sweating, slower rate of acclimatisation to heat, and impaired thirst and fluid ingestion responses [36]. These events can be aggravated by high ambient temperature via increased heat stress and reduced availability of clean drinking water, especially in impoverished conditions where there is poor access to rehydration solutions from primary health facilities. Children with co-morbidity of severe malnutrition may have increased vulnerability to dehydration or other serious complications [37,38]. This could explain the immediacy of mortality risks at high temperatures but not for the hospitalised severe cases in which other pathways may be involved given the peak risk at lag week 3.

The findings for hospitalised severe diarrhoea cases have some uncertainties given the sensitivity of the results when certain model specifications were altered. This may be due to the low number of cases from districts other than the Capital District that make the modelling unstable. Moreover, we only analysed severe cases from a single hospital, which may not be truly representative of severe cases in the NCR districts. Adding more hospitals may help further elucidate the association between ambient temperature and severe diarrhoea.

There were several limitations in the study. First, we were not able to conduct any stratified analysis for aetiology to confirm aetiology-specific temperature associations for both mortality and hospital admissions since the majority were pathogen-unspecific diarrhoea. This is due to the routine diagnostic tests for diarrhoea such as stool culture and microscopy in hospitals and laboratories in developing countries such as the Philippines, having a long turn-around time for slow-growing pathogens and can result in false negatives [39,40]. It may also be difficult for a physician to link a cause of death on any specific enteropathogen given the limitations of routine diagnostic tests and in certain instances during verbal autopsy. Second, there may be possible inaccuracies in assigning diarrhoea as the cause of death, particularly the ones that took place outside the hospital [41,42]. We were not able to separately analyse the deaths with verbal autopsy or the ones not attended by any physician since they are low in number. Lastly, the current study that analysed the data from three urban districts in the Philippines may have limited generalizability, which can be improved through a wider coverage.

With the projected warming of climate [43], extreme heat events may occur more frequently and intensely [44] that may greatly impact severe diarrhoea cases. Health adaptation in the future should include strengthening the proper management of dehydration through effective delivery of oral rehydration solution and zinc supplementation services from primary healthcare facilities. Furthermore, sustaining the gains from water, sanitation, and hygiene (WASH), as well as child nutrition programmes can also augment the efforts to prevent infections and severe cases [45].

## 5. Conclusions

Severe diarrhoea cases were associated with ambient temperature and their risks were found to increase under high- and low-temperatures in the three urban districts in the Philippines. Moreover, the risks were greater at high temperatures than at low temperatures, especially for mortality due to diarrhoea. Given the projected ambient temperature rise in the future, primary healthcare services may need to be strengthened to address the possible rise in diarrhoea and dehydration, as well as sustain the gains in water, sanitation, and hygiene particularly in poor communities to prevent diarrhoeal diseases.

## Figures and Tables

**Figure 1 ijerph-18-08191-f001:**
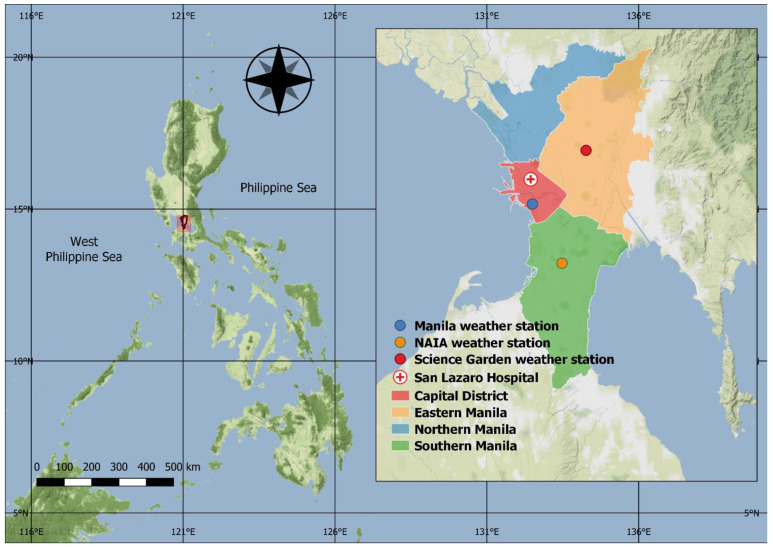
National Capital Region districts, weather stations, and San Lazaro Hospital. Map is generated using QGIS 3.16 with stamen.

**Figure 2 ijerph-18-08191-f002:**
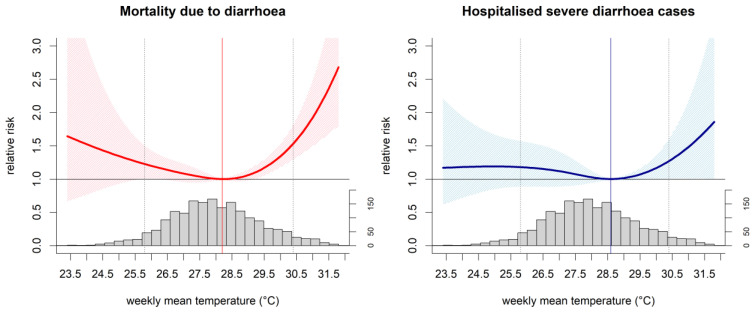
Cumulative associations between weekly mean temperature and diarrhoea in the National Capital Region in 2006–2017. Curves are the relative risks, polygons are the 95% confidence intervals, vertical lines are the minimum risk temperatures, grey dotted lines are 5th and 95th temperature percentiles, and grey histograms are the temperature distributions.

**Table 1 ijerph-18-08191-t001:** Summary statistics of weekly study variables in the National Capital Region of the Philippines, 2006–2017.

Location	Weekly Mean Values (Range: Minimum, Maximum)			
	Diarrhoeal Deaths	Hospitalised Severe Diarrhoea Cases *	Mean Temperature (°C)	Dew Point Temperature (°C)
Capital District	1.9 (0, 10)	6.1 (0, 32)	28.5 (24.5, 31.8)	23.6 (17.8, 26.4)
Eastern Manila District	3.2 (0, 17)	1.4 (0, 22)	27.4 (23.3, 31.3)	23.1 (18.2, 25.7)
Southern Manila District	3.2 (0, 10)	0.9 (0, 12)	28.1 (24.2, 31.8)	23.9 (17.4, 27.4)
National Capital Region **	8.3 (0, 37)	9.1 (0, 50)	28.0 (24.1, 31.6)	23.5 (17.8, 26.2)

* Hospitalised severe diarrhoea cases from San Lazaro Hospital; ** average of three districts.

**Table 2 ijerph-18-08191-t002:** Relative risks (RR) and 95% confidence intervals (CI) for the association of severe diarrhoea with high temperatures (HT) and low temperatures (LT) relative to the minimum risk temperatures (MRT) in the National Capital Region of the Philippines in 2006–2017.

Location	Temperatures (°C)		Mortality			Hospitalised Severe Diarrhoea		
	95% Pctl	5% Pctl	MRT (°C)	RRs (95% CI) for HT	RRs (95% CI) for LT	MRT (°C)	RRs (95% CI) for HT	RRs (95% CI) for LT
Capital District	30.6	26.4	28.9	1.540 (1.303; 1.820)	1.139 (0.859; 1.511)	29.0	1.267 (0.979; 1.640)	1.166 (0.831; 1.637)
Eastern Manila District	29.8	25.2	27.5	1.504 (1.261; 1.794)	1.342 (0.981; 1.836)	28.2	1.284 (1.014; 1.625)	1.263 (0.870; 1.833)
Southern Manila District	30.6	26.1	28.2	1.646 (1.351; 2.005)	1.319 (1.004; 1.735)	28.6	1.270 (0.937; 1.722)	1.113 (0.784; 1.581)
National Capital Region	30.4	25.8	28.2	1.533 (1.294; 1.817)	1.226 (1.004; 1.496)	28.6	1.271 (1.007; 1.604)	1.179 (0.886; 1.57)

## Data Availability

The dataset and R codes are available in the P.L.C.C. GitHub repository (https://github.com/paulcarlos, accessed on 30 July 2021).

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
