# Peer review of "Association between Ambient Temperature and Severe Diarrhoea in the National Capital Region, Philippines"

_ijerph, 2021, doi:10.3390/ijerph18158191_

Round 1
Reviewer 1 Report
REVIEW - "Association between Ambient Temperature and Severe Diarrhoea in the National Capital Region, Philippines"
The article untitled "Association between Ambient Temperature and Severe Diarrhoea in the National Capital Region, Philippines" has a very good methodological sector which embases a short but very robust discussion sector. That is a well developed and interesting research. Nonetheless, there is always a very difficult theme to work: could be diarrhoea cases explained through a environmental/deterministic bias instead of a social and economical one? Anyway, I realized that this work is a very elegant one, which has good conditions to transit between these two very delicated poles. There are only few observations:
Lines 111 - 114: "We estimated the associations between weekly average
temperatures and severe diarrhoea outcomes by fitting generalised linear model in combination with a distributed lag nonlinear model to model the non-linear and possibly delayed effects of temperature [17]". - This part of the text is being a bit confused to the reader.
REFERENCES: Please check and organize the structure of the references sector.
Thank you, that is a very good research in my opinion.
Author Response
We thank the reviewer for her/his time and the positive feedback on our work. We do agree that socioeconomic determinants are important in modelling diarrhoea and should be incorporated with environmental determinants. We did try to consider some socioeconomic determinants in the meta-regression but only population density was found to be a significant meta-predictor reducing the heterogeneity for diarrhoea mortality. We do believe future research should consider this.
For the specific comments, we edited the said sentence to be less confusing to the readers and edited each reference to complete their information.
Page 5, line 146-148: “…we estimated the associations between weekly average temperatures and weekly severe diarrhoea outcomes by fitting generalised linear model in combination with a distributed lag nonlinear model…”
Reviewer 2 Report
The findings of the study are not surprising and novelty. the positive association of ambient temperature and incidence of diarrhoea is well-known altogether so parallelly severe diarrhoea cases including deaths is nothing new except one increased risk of diarrhoea at low temperature. Authors should write more why? It is a real-life ( fact) or statistical coincidence? Explanation of abbreviations used in equation 9 lines 118-125) will be easier to get in form of a list instead of text.
Author Response
We thank the reviewer for her/his time and feedback on our work. The increased risk at low temperatures may be explained by viral infections like rotavirus which was found to have negative associations with temperature. We added a phrase to highlight this in the discussion. For the equation in page 5 line 153, we have decided to maintain the current version to be consistent with the common format for explaining equation in similar research.
Page 8, line 285-288: “The temperature associations found may be explained by the aetiologies that comprise the severe diarrhoea cases. Under high temperatures, risks for shigellosis, E. coli enteritis, salmonellosis, and cryptosporidiosis (including other bacterial enteropathogens) increase while risks for rotaviral and noroviral enteritis increase under low temperatures…”
Reviewer 3 Report
Association between Ambient Temperature and Severe Diarrhoea in the National Capital Region, Philippines
General Comment: Paper considers an interesting topic for research with full justification. The presentation is coherent and inclusive, however, there are few grammatical issues of language which would disappear by a thorough read.
Abstract: The first sentence of the abstract is misleading and may go against the theme of the paper. It is because authors posit that the link between ambient temperature and diarrhea are well characterize then what differentiates severe diarrhea from the former? However, the rest of the abstract is quite clear and understandable. Similarly, the last sentence is also confusing as to what means from ‘gains in water……..’
Introduction: It is well-connected and well-founded and builds the case effectively.
Methods: Methods are well-described and aptly used although in abstract, it is mentioned that two-stage time series analysis was conducted while in the section on methods, no such information about two stage time series analysis given.
Discussion: Authors have profoundly discussed the findings and given implications as well as limitation of the work. The conclusions are well-drawn and based on the study findings.
Minor: First sentence of introduction has a grammatical error.
Author Response
We thank the reviewer for her/his time and the positive feedback on our work. We edited the abstract as follows:
Page 2, line 35-37: “Epidemiological studies have quantified the associations between ambient temperature and diarrhoea, but, to our knowledge, no study has quantified the temperature associations for severe diarrhoea cases…”
Page 2, line 52-53: “…and sustain the improvements made in water, sanitation, and hygiene…”
We added a sentence and phrases to show the two-stage approach:
Page 5, line 145-148: “We applied a two-stage time series approach to model the non-linear temperature associations of severe diarrhoea outcomes. In the first stage, we estimated the associations between weekly average temperatures and weekly severe diarrhoea outcomes by fitting generalised linear model in combination with a distributed lag nonlinear model”
Page 5, line 170-171: “In the second stage, we used an intercept-only meta-regression model to combine the district-specific estimates”
Finally, we have also tried to improve the grammar of the introduction as suggested in page 3 line 61.
Reviewer 4 Report
Ambient Temperature and Severe Diarrhoea
This is a very interesting report on ambient temperature and severe diarrhoea. It is well designed with extensive data collection. The analysis is expert, thoughtful, and well performed. The results are interesting. Below are a few minor suggestions.
The mean risk temperature is difficult to understand in the Abstract until one has read the article. As I read the Abstract, I assumed that the MRT would be the lowest temperature, not approximately the average temperature. Also reading the Abstract, I was immediately curious what the 5th and 95th temperatures were. All of this could be solved if these temperatures were actually reported in the Abstract. The NCR 5th percentile temperature was 25.8 and the 95th percentile temperature was 30.4. The MRT for mortality was 28.2 and for severe diarrhoea was 28.6.
The relative risk from the 95th and 5th percentiles from the splines are interesting and worth reporting. However, they are really point estimates from a calculated abstraction. Wouldn’t it be more interesting to find the RR for diarrhea above a fixed temp such as 30 degrees. And below 25. This would be more practical and could translate more easily to other locations. For example, public health officials in Kenya could more easily apply a RR for diarrheoa at temperatures above 30 degrees compared to a point estimate at 30.4 degrees. I suggest an additional table with this information.
The authors hinted at a possible interesting subgroup analysis. They reported that diarrheoa afflicted both the very young and the very old. This immediately raises the question of whether the J-shaped curve would show in both age sub-groups.
Author Response
We thank the reviewer for her/his time and the positive feedback on our work. We edited the abstract to add the temperature information:
Page 2, line 42-43: “…We computed the relative risks at the 95th (30.4°C) and 5th percentiles (25.8°C) of temperatures…”
Page 2, line 46-50: “Mortality risks were found to increase by 53.3% [95% confidence interval (CI): 29.4%; 81.7%)] at 95th percentile of weekly mean temperatures compared with the MRT (28.2°C). Similarly, the risk of hospitalised severe diarrhoea increased by 27.1% (95%CI: 0.7%; 60.4%) at 95th percentile in mean weekly temperature compared with the MRT (28.6°C).”
For the presentation of relative risks, we chose to present the risks in specific temperatures (95th and 5th percentiles) relative to the minimum risk temperatures in order to be consistent with the standard way of presenting the risk for temperature exposure based on the modelling approach by Gasparrini 2011 (doi: 10.18637/jss.v043.i08) which has been widely used in epidemiological studies of nonlinear temperature exposure. This consistency is helpful for comparison across locations. Given the nonlinear exposure-response function based on the modelling approach, instead of presenting multiple effect estimates under a set of absolute temperatures, we refer the reader to the estimated relative risk curves across the temperature range in Figure 2 to illustrate the overall association at the study location. Lastly, we were not able to perform subgroup analysis for the very young children and the very old because there were not enough cases to model.